# The Antioxidant Effect of Dietary Bioactives Arises from the Interplay between the Physiology of the Host and the Gut Microbiota: Involvement of Short-Chain Fatty Acids

**DOI:** 10.3390/antiox12051073

**Published:** 2023-05-10

**Authors:** Rossana Cuciniello, Francesco Di Meo, Stefania Filosa, Stefania Crispi, Paolo Bergamo

**Affiliations:** 1Institute of Biosciences and BioResources-UOS Naples CNR, Via P. Castellino, 111-80131 Naples, Italy; 2IRCCS Neuromed, 86077 Pozzilli, Italy; 3Department of Medicine, Indiana University, Indianapolis, IN 46202, USA

**Keywords:** MACs, polyphenols, PUFAs, conjugated linoleic acid, gut microbiota, active metabolites

## Abstract

The maintenance of redox homeostasis is associated with a healthy status while the disruption of this mechanism leads to the development of various pathological conditions. Bioactive molecules such as carbohydrates accessible to the microbiota (MACs), polyphenols, and polyunsaturated fatty acids (PUFAs) are food components best characterized for their beneficial effect on human health. In particular, increasing evidence suggests that their antioxidant ability is involved in the prevention of several human diseases. Some experimental data indicate that the activation of the nuclear factor 2-related erythroid 2 (Nrf2) pathway—the key mechanism in the maintenance of redox homeostasis—is involved in the beneficial effects exerted by the intake of PUFAs and polyphenols. However, it is known that the latter must be metabolized before becoming active and that the intestinal microbiota play a key role in the biotransformation of some ingested food components. In addition, recent studies, indicating the efficacy of the MACs, polyphenols, and PUFAs in increasing the microbial population with the ability to yield biologically active metabolites (e.g., polyphenol metabolites, short-chain fatty acids (SCFAs)), support the hypothesis that these factors are responsible for the antioxidant action on the physiology of the host. The underlying mechanisms through which MACs, polyphenols, and PUFAs might influence the redox status have not been fully elucidated, but based on the efficacy of SCFAs as Nrf2 activators, their contribution to the antioxidant efficacy of dietary bioactives cannot be excluded. In this review, we aimed to summarize the main mechanisms through which MACs, polyphenols, and PUFAs can modulate the host’s redox homeostasis through their ability to directly or indirectly activate the Nrf2 pathway. We discuss their probiotic effects and the role played by the alteration of the metabolism/composition of the gut microbiota in the generation of potential Nrf2-ligands (e.g., SCFAs) in the host’s redox homeostasis.

## 1. Introduction

Oxidative eustress represents the physiological exposure to low doses of endogenous oxidant species, produced by cells to address specific targets via the redox network to maintain cell homeostasis [1]. The alteration of the redox status is largely involved in cells and consequent organ dysfunction, which often leads to a wide variety of chronic and age-related human diseases. One of the molecular pathways responsible for the preservation of this equilibrium state is the Nrf2/Keap1 pathway. Among several pathways regulated by Nrf2, there is the maintenance of proteostasis, whose dysfunction determines cell death by autophagy or apoptosis caused by events such as protein misfolding and aggregation [2].

The intestine is an essential organ involved in human nutrition, and increasing evidence indicates that the interplay between gut commensal bacteria (microbiota)—the complex microbial community that colonizes the human gut—and its composition is influenced by the host’s genotype, environment, and diet. In particular, food nutrients play a key role in human metabolism and health via the modulation of multiple mechanisms, including energy metabolism, intestinal homeostasis, antioxidant homeostasis, and immune responses [3]. In particular, the metabolic activity of gut microbes is essential for maintaining host health, and alterations in its composition induce metabolic shifts that may have adverse effects. Under healthy conditions, the preponderance of potentially beneficial bacterial species such as *Firmicutes* and *Bacteroides* over potentially pathogenic ones such as *Proteobacteria* is called eubiosis, and it has been associated with a healthy status of the host organism. On the contrary, the perturbation or disruption of this composition, known as dysbiosis, has been associated with several metabolic or immune disorders [4].

The consensus on microbiota-mediated healthy effects on the host is based on the microbe-induced biotransformation of food components into bioactive metabolites. Bioactive molecules exhibit, in combination with food components, the ability to modulate the metabolic pathways of the host or to modify the composition and metabolism of the microbiota. Among them, non-digestible fibers, also known as microbiota-accessible carbohydrates (MACs), polyphenols, and PUFAs, are the best-characterized food components influencing the composition and metabolism of the microbiota [5].

Analogously, the regulation/activation of the Nrf2 pathway can be exerted by endogenous ROS (Reactive Oxygen Species) or exogenous molecules, such as dietary bioactive molecules that may need to undergo structural transformations by the gut microbiota before performing their bioactivity. The link between Nrf2 and gut microbiota health may be hypothesized on the basis of the association of Nrf2 dysfunction with the alteration of the composition of the microbiota due to ageing or pathological conditions. Unfortunately, there is only one study that supports this connection [6] and, as far as we know, the molecules responsible for this crosstalk have not yet been indicated.

In this review, we first discuss the role of microbial by-products, such as short-chain fatty acids (SCFAs), on Nrf2-mediated oxidoreductive homeostasis (redox status). Then, we provide a comprehensive and updated overview of the interplay of MACs, polyphenols, and PUFAs on the composition/metabolism of the gut microbiota and their direct action on specific molecular targets or their indirect action—via the modulation of the composition of the microbiota—which, downstream, modulates the host’s redox status via the production of SCFAs (Figure 1).

## 2. Importance of the Nrf2 Pathway and Its Link with Gut Microbiota

Nrf2 is a redox-sensitive transcription factor, and it is the master regulator of oxidoreductive and immune homeostasis. In the cell cytoplasm, Nrf2 is associated with the inhibitory protein Kelch-such as ECH-associated Protein 1(Keap1), which, owing to the presence of specific cysteine residues, acts as a sensor of endogenous and exogenous prooxidants [7]. In particular, under a mild increase in oxidative stress, specific cysteine residues in Keap1 allow newly synthesized Nrf2 to escape Keap1-mediated ubiquitination and to activate the transcription of Nrf2-target genes (more than 200) involved in fundamental biochemical pathways (e.g., mitochondrial functions, oxidoreductive and immune homeostasis). Due to its pleiotropic activity, Nrf2 has been indicated as a therapeutical target for a variety of human diseases [8], and the activation of this molecular mechanism has been mainly involved in the cytoprotective activity of dietary antioxidants, including plant polyphenols [9], dietary PUFAs, and some of their metabolites [10].

As for SCFAs, bioactive polyphenol metabolites can affect specific pathways determining the modulation of specific target genes. In particular, their antioxidant activity is achieved through the activation of the Nrf2 pathway [11]. However, based on their low bioavailability and their biotransformation by the intestinal microbiota, it is probable that their antioxidant ability is mediated by some of their metabolites or by other molecules of bacterial origin (e.g., SCFAs). These fatty acids may be at the crossroad between the diet and the organism’s redox status.

This hypothesis is consistent with a recent result showing the positive correlation between Nrf2 activation in the brain with the levels of SCFA-producing bacteria (e.g., *Roseburia*, *Oscillibacter*, *Faecalibaculum*) in mice treated with several Nrf2 activators [6]. In this framework, an overview of the modulatory effects of MACs, PUFAs, and polyphenols to increase the level of SCFA-producing bacteria along with data reporting the efficacy of SCFAs as Nrf2 ligands will be summarized in the following sections.

## 3. Composition of the Gut Microbiota

The gut microbiota comprise thousands of bacterial species, mainly those of *Bacteroidetes* (9–42%), *Firmicutes* (30–52%), and *Actinobacteria* (1–13%) [12,13], whose composition is influenced by the environment and host genotype, as well as by age and diet. In adults, bacterial cells can reach gut concentrations of up to 10^14^ cells, representing the largest number and highest concentration of microorganisms found in the human body [14]. The great variety of bacterial species characterizing the microbiota results in the expression of a large number of genes. It has been estimated that the whole genomic content of gut microbiota exceeds that of humans by one hundred times, suggesting that the genome of the microbiota displays a metabolic potential capacity to influence the physiology of the host [15]. Recent advances in high-throughput sequencing technologies have allowed us to easily identify the genomes of ecosystem samples, contributing to the comprehension of the role of the gut microbiome in health and disease (Human Microbiome Project) [16].

The homeostatic condition of the intestinal microbiota (eubiosis) mainly depends on the balanced diversity of these microbial populations and the controlled growth of potentially pathogenic bacteria. A reduction in the diversity of the intestinal population (dysbiosis) can directly affect the epithelial and mucosal functions, leading to an inflammatory environment in the gastrointestinal tract [17] as well as other human pathologies such as non-alcoholic fatty liver disease and neurodegenerative disorders. The activity of the gut microbiota is essential in the host metabolism, protecting against infections from pathogens and intervening in energy homeostasis and the immune response by coordinating specific gene expression in response to different host and environmental signals [18]. In particular, the *Firmicutes* to *Bacteroidetes* ratio (F/B ratio) has been extensively examined in the human and mouse gut microbiota, and it has been demonstrated that the F/B ratio is associated with metabolic diseases [19], inflammatory diseases [20], neuropsychiatric disorders, and cancer [21,22,23].

## 4. Effect of SCFAs on the Composition of the Gut Microbiota

The beneficial effects associated with the diversity of the microbial population arise from the metabolic activities of specific microbial populations. Under eubiotic conditions, the commensal relationship between the microbiota and the host mainly consists of the capacity of bacteria to generate bioactive metabolites, starting from the ingested food, which exhibits the ability to modulate different metabolic pathways of the host [24]. For example, the production of carboxylic acids with aliphatic tails with fewer than six carbon atoms such as acetate (C2), propionate (C3), and butyrate (C4), resulting from the anaerobic fermentation of dietary plant polysaccharides, is the most relevant metabolic activity of enteric microbiota. These molecules are collectively referred to as Short-chain Fatty Acids (SCFAs) [25].

The growth of anaerobic SCFA-producing bacteria is favored by the low oxygen concentrations in the intestine where the two most abundant populations, namely, *Bacteroidetes* and *Firmicutes*, mainly produce acetate/propionate and butyrate, respectively [26]. Interestingly, due to butyrate generation during acetate metabolism, their coexistence can be consequential to mutual metabolic gain, thus resulting from the utilization of acetate produced by *Bacteroidetes* and *Firmicutes* to produce butyrate and propionate [27]. This example strongly supports the concept that the production of SCFAs is finely tuned by the balance of the bacterial species present in the gut.

The homeostatic condition of the intestinal microbiota can be restored by the level of SCFAs, and many studies in vivo describe the link between gut dysbiosis and the production of SCFAs (Table 1).

It can be assumed that butyrate, being a fundamental nutrient for colonocytes, satisfies the metabolic demands of the colon epithelium [42], and it also modulates the expression of tight junction proteins, thus preserving the intestinal barrier whose integrity is a crucial part of the overall immune response [43]. In addition, local O_2_ consumption during butyrate uptake and its metabolism by the intestinal epithelium stabilizes the hypoxia-inducible factor (HIF)—a transcription factor that coordinates barrier protection—which promotes the creation of an anaerobic environment. This “physiological hypoxia” stimulates the growth of SCFA-producers (anaerobic bacteria) [44], indirectly regulating the functionality of the intestinal barrier [45]. In addition, SCFAs have been shown to display an inhibitory effect on the growth of potentially pathogenic bacteria such as *Salmonella typhimurium* [46] or *Clostridium difficile* [47].

### 4.1. Effect of SCFAs on Gut Homeostasis

Acetate, propionate, and butyrate in the colon are present in the molar ratio 60:25:15, although proportions can vary depending on factors such as diet, microbiota composition, the site of fermentation, and the genotype of the host [48]. These are the predominant SCFAs present in the proximal regions of the large intestine in humans and rodents, and they are present at mM levels [49,50,51].

Acetate, propionate, and butyrate reach the highest concentrations (70–140 mM) in the proximal colon [48] where they enhance mucin secretion by increasing the expression of the MUC2 gene [52], with a concentration gradient decreasing from the lumen to the periphery [53]. When these SCFAs are absorbed into hepatic portal circulation and the lacteal lymphatic system, they reach total concentrations ranging from 375 μM to 148 μM in the portal and hepatic blood respectively, or 79 μM in peripheral blood [48,54]. Butyrate and propionate, mostly metabolized by hepatocytes, were reported in a range of 1–15 μM in the systemic circulation, while acetate ranged between 100 and 200 μM [55,56]. However, the small amounts of SCFAs present in the bloodstream are sufficient to elicit a wide range of biological functions in different tissues.

A study on a mouse model of induced colitis demonstrated that SCFAs preserve gut homeostasis by acting on the inflammasome pathway through the upregulation of interleukin (IL)-18 [57]. Accordingly, low levels of butyrate and propionate-producing bacteria were found in patients suffering from inflammatory bowel diseases (IBD) such as ulcerative colitis or Crohn’s disease [31,32]. Several in vivo analyses have indicated that SCFAs regulate gut motility by stimulating mucosal receptors [58] or by increasing the release of the Peptide YY from gut endocrine cells, thus favoring intestine motility [59]. Other studies demonstrated that SCFAs act in preventing colonic diseases, by enhancing the absorption of minerals and decreasing the cholesterol concentration [60,61]. Experiments using germ-free animals reported that their reduced gut motility can be restored by the infusion of SCFAs [62].

Convincing evidence supports the idea that the beneficial effect of SCFAs extends beyond the colon. In fact, SCFAs participate in different physiological processes in the human body, being able to improve gut physiology, modulate the host’s glucose and lipid metabolism, and affect immune function [63,64]. In particular, upon their transport from the intestinal lumen into the blood compartment of the host, SCFAs are absorbed by the liver for gluconeogenesis or by muscle to generate energy [65]. Among SCFAs, acetate is the primary substrate for cholesterol synthesis [66], while propionate inhibits cholesterol synthesis by reducing serum lipids and has a protective effect against colon cancer [64,67]. Notably, SCFAs modulate brain functions by acting on the production of neuroactive metabolites [68]. For example, butyrate and propionate can be transferred from the gut to the brain where they act as signalling molecules through the monocarboxylate transporters that are highly expressed in the blood–brain barrier [69]. Finally, butyrate has been reported to play a protective role against carcinogenesis in colon cancer cells by enhancing the expression of cell cycle inhibitory genes [70].

### 4.2. Signaling Mechanisms Induced by SCFAs

The detailed description of the mechanisms involved in the signalling activity of SCFAs does not fall within the scope of this work and, due to its complexity, only a synthetic presentation is reported herein. Besides the relevant role in intestinal health, SCFAs may play their signalling role via the activation of several biochemical pathways: G-protein-coupled receptors (GPCRs), histone deacetylases (HDACs), and Nrf2 [71,72,73] (Figure 2).

In humans, there are at least six GPRs that are sensitive to SCFAs, but among them, only GPR41, GPR43, and GPR109A are involved in SCFA-mediated signaling. GPR41 and GPR43 are the best-studied SCFA receptors [74] and are activated by acetate, propionate, and, to a lesser extent, also by butyrate. GPR41 is expressed in colon cells, in the blood vessels, and in the sympathetic nervous system, while GPR43 is mainly expressed in enteroendocrine L cells, lymphocytes, neutrophils, and monocytes [75]. GPR109A has a high affinity for niacin that can be activated by butyrate, and it is expressed only in human immune cells and colonocytes. In addition, GPR109A is highly expressed in adipocytes. The activation of this receptor in adipocytes has been linked to lipolysis and a decrease in plasma free fatty acids [76]. Activated GPCR receptors can regulate different signaling via the activation of many cellular functions such as the mitogen-activated protein kinase (MAPK) family of serine-threonine kinases, including extracellular signal-regulated kinase (ERK), c-jun N-terminal kinase (JNK), p38, and ERK5, through an intricate network of signaling. The activation of GPR43 also stimulates the phospholipase-C determining intracellular Ca^2+^ release and the activation of protein kinase C [77].

HDACs are a group of enzymes that affect gene transcription or alter protein activity by removing the acetyl group on the lysine ϵ-amino group of the target protein. The inhibition of HDACs is relevant for immune and inflammatory regulation by modulating either innate immunity through regulation of the Toll-Like Receptor (TLR) and Interferon (IFN) signaling pathways or by regulating antigen presentation and B and T lymphocytes to achieve adaptive immunity [78,79]. In particular, the inhibitory effect of HDACs on SCFAs, mainly due to propionate and butyrate, results in an anti-inflammatory effect through the promotion of regulatory T cell (Treg) development as well as CD4^+^ T cell IL-10 production [80,81].

Interestingly, the functional link existing between this signaling pathway and gut microbiota homeostasis has been indicated by (a) the modulatory ability of SCFAs in the Nrf2 pathway [72], (b) the age-dependent decline in the concentration of SCFAs in the gut [82], and (c) the positive association between microbiota diversity and Nrf2 efficacy [83]. In addition, the link between the production of SCFAs and the Nrf2 pathway was indicated in a recent study showing the ability of *Clostridium butyricum* pretreatment to increase the SCFA contents in the cecum of Enterotoxigenic *Escherichia coli* K88 (ETEC K88)-infected mice. In particular, the data indicated that such improvement was associated with the amelioration of the oxidative damage induced by ETEC K88 infection through the activation of the Nrf2 pathway [84]. A summary of the differential ability of microbial SCFAs in activating different receptors involved in the Nrf2 pathway is shown in Table 2.

Finally, the interplay existing between different SCFAs further strengthens the complexity of their mechanism of action [85,86] (Figure 2).

## 5. Dietary Bioactive Molecules and the Gut Microbiota Composition

Diet has a fundamental role in determining the composition of the gut microbiota. MACs, PUFAs, and polyphenols are the most well-characterized food components able to modulate the composition and metabolism of microbiota. All of these food components can exert prebiotic effects that result in the modulation of bacterial strains producing SCFAs. These compounds can affect specific molecular mechanisms that result directly in beneficial effects on the host’s health or indirectly allow the gut microbiota to produce active/antioxidant metabolites. A detailed description of the complex network involved in the SCFA-mediated signaling activity and its therapeutic relevance have been presented elsewhere by [87] and will not be further examined herein.

### 5.1. Effect of Microbiota-Accessible Carbohydrates (MACs) on Gut Homeostasis

Non-digestible polysaccharides such as resistant starch, inulin, cellulose, guar gum, and pectin—collectively known as MACs—are the main energetic source of gut bacteria. MACs exert a modulatory role in the gut microbial composition, maintaining gut homeostasis. An MAC-rich diet in humans is associated with an increased content of colonic and fecal SCFAs (Table 3). On the contrary, a high-fat and high-sucrose diet can determine dysbiosis onset, which represents the first step in the development of an increased susceptibility to inflammatory diseases such as IBD or non-alcoholic fatty liver disease or colon cancer [88].

After ingestion, intact MACs reach the colon where they are metabolized by microbial enzymes, i.e., glycoside hydrolases and polysaccharide lyases. These enzymes, degrading the glycosidic bonds, convert MACs into monosaccharides [89]. The digestion of MACs by gut bacteria yields SCFAs that can be easily absorbed by gut epithelial cells, exerting healthy effects on human health [90]. It is important to underline that food intake has a crucial role in the fine-tuning of SCFA production by gut bacterial species. This implies that dietary changes affecting the MAC content can exert a prebiotic effect, altering the composition and metabolism of the gut microbiota. As previously mentioned, the eubiotic condition is associated with “good health,” while low microbial diversity and dysbiosis have been correlated with diseases highly prevalent in Western society, such as obesity, type 2 diabetes, or IBD [91]. Several studies have shown a positive correlation between the intake of a vegetable diet enriched in fiber and microbiota diversity and the enrichment of SCFA-producing bacteria in human populations [92,93].

**Table 3 antioxidants-12-01073-t003:** The effect of MACs on the composition of microbiota and the production of SCFAs. An increase or decrease in the levels considered is indicated by upward (↑) or downward arrow (↓), respectively.

Treatment	Model	Microbiota Alteration/SCFA Production	Ref.
**Metanalysis**	Studies investigating the effect of dietary fiber on gut microbiota	↑ *Bifidobacterium* ↑ SCFAs	[29]
**Dietary Fiber**	European children (Low-fiber diet) vs. Rural African village (High-fiber diet)	↑ *Bacteroidetes* ↑ SCFAs	[92]
	African Americans vs. Rural native Africans	↑ *Prevotella* ↑ SCFAs	[94]
**Inulin**	Mice with hyperuricemia vs. wild-type mice	↑ microbial diversity ↑ SCFA-producing bacteria (*Akkermansia* and *Ruminococcus*). ↑ acetate, propionate, and butyrate	[95]
	Nonalcoholic Fatty Liver Disease rat model	↑ *Bifidobacterium*, *Phascolarctobacterium*, *Blautia* ↓ Acetate ↑ Propionate and Butyrate	[96]

### 5.2. Effect of Polyphenols on Gut Homeostasis

Polyphenols are dietary bioactive compounds derived from plants and present in fruits and vegetables. These compounds are chemically characterized by the presence of at least one phenyl ring and one or more hydroxyl substituents. Dietary polyphenols are characterized by poor absorption and extensive metabolism [97].

Most of the dietary-ingested polyphenols are present as glycosylated ester or polymer forms, and after ingestion, they are recognized as xenobiotics. Their structural complexity determines that they reach the large intestine without modifications. In the gut, polyphenols—through microbic metabolism—are converted into low-molecular-weight metabolites and then absorbed from epithelial cells to reach the plasma.

Small polyphenols can be easily absorbed after de-glycosylation by bacterial enzymes. Then, they are converted into soluble metabolites through Phase I (oxidation, reduction, and hydrolysis) and Phase II reactions (conjugation) [98]. On the contrary, complex polyphenols (oligomeric and polymeric) need to be transformed by specific gut enzymes to increase their bioavailability and plasma levels. As glycosides, polyphenols are first converted into aglycones through specific enzymatic transformations. Subsequently, they undergo additional modifications including C-ring cleavage, decarboxylation, dehydroxylation, and demethylation [99]. The released aglycones, after absorption into the small intestine, can be further metabolized by enterocytes and hepatocytes. In the liver cells, polyphenols undergo Phase II biotransformation with the production of glucuronide and sulfation metabolites, followed by distribution to organs and excretion in urine [100]. As a consequence, the final metabolites are quite different from the parent molecules present in the ingested food. In addition, their residence time in the plasma is extended compared to parental compounds, thus allowing them to exert biological effects [101].

Polyphenols are present in low concentrations in foods, as compared to macronutrients, and their low bioavailability results in a lower amount absorbed by the body. The strong discrepancy between the biological activity of polyphenols and their concentration has led to the formulation of the low bioavailability/high bioactivity paradox [102,103].

The biotransformation of polyphenols by gut microbiota and the capacity of polyphenols to modulate gut microbiota have been reported [104,105]. Consequently, less than 5% of the total polyphenolic intake is absorbed and reaches the plasma unchanged while their microbial metabolites predominate in the plasma [106].

For example, resveratrol, widely distributed in grapes, berries, and peanuts, is transformed by the gut microbiota into different metabolites: dihydroresveratrol (DHR), which is partly absorbed and further metabolized to two conjugated forms: monosulfate (DHR) and monoglucuronide (DHR). An analysis of the bioavailability of these microbial metabolites indicates that upon daily intake of 500 mg of pure trans-resveratrol, the metabolite concentration in plasma increased from 3 to 13 μM [107].

Curcumin, found in turmeric (*Curcuma longa* (Linn.)), is converted into active metabolites in a two-step reaction: the first produces dihydro curcumin (DHC) from curcumin, and then DHC is converted into tetrahydro curcumin (THC). It has been reported that supplementation with 1 g of turmeric acid and curcumin was not detectable in plasma while the concentration of the active metabolites persisted in blood for at least 8 h, ranging from 2 to 47 nM [108].

Bioactive polyphenol metabolites can affect specific pathways determining the modulation of specific target genes. Quercetin, a glycoside present in many fruits and vegetables, is transformed into active glucuronated, sulfated, and methylated metabolites in the enterocytes of the small intestine. Quercetin that is not absorbed is metabolized by specific bacterial enzymes able to cleave the quercetin C-ring, producing DOPAC (3,4-Dihydroxyphenylacetic acid) PCA (3,4-dihydroxybenzoic acid), and 3-OPAC (3-hydroxyphenyl acetic acid). These metabolites may exert their effect by directly modulating different signaling pathways (Figure 3), such as the activation of the Nrf2 pathway that is involved in their antioxidant activity [11].

As for SCFAs, bioactive polyphenol metabolites can affect specific pathways determining the modulation of specific target genes. For example, their antioxidant activity is achieved through the activation of the Nrf2 pathway. Several in vitro and in vivo studies demonstrated that polyphenols can activate the Nrf2 pathway [11]; however, owing to their low bioavailability and their bio-transformation by intestinal microbiota, whether polyphenols exert their modulatory effect directly or indirectly—through their metabolites—remains to be established.

Different microbial genera including *Bacteroides*, *Enterococcus,* and *Eubacteria* play a key role in determining the metabolic fate of polyphenols [109]. In addition, some polyphenol bio-transformations require the presence of specific bacteria. For example, gut bacteria involved in resveratrol metabolism are *Slackia equolifaciens* and *Adlercreutzia equolifaciens*. Curcumin is converted into active metabolites mainly by *Escherichia coli*, but it can also be metabolized by *Blautia* sp. [110], and quercetin can be converted into active metabolites by specific bacterial strains such as *Eubacterium ramulus*, *Clostridium orbiscindens*, *Eubacterium oxidoreducens*, and *Butyrovibrio* spp. that can cleave the C-ring [111]. The importance of gut bacteria for polyphenol metabolism is further indicated by the fact that germ-free animals are not able to produce phenolic metabolites [112]. However, individual differences in gut microbiota composition can account for individual variations in absorption, metabolism, and polyphenol bioactivity. Thus, the identification of bacterial species responsible for polyphenol metabolism is crucial to unraveling the health-promoting effects on the host [113]. Individual differences in microbiota compositions can influence the metabolic fate of ingested polyphenols [114]. This can explain the reason why a similar daily intake of polyphenols results in different health effects.

In addition, their conversion into bioactive metabolites can have beneficial effects on the host’s health. In particular, the intestinal microbiota conversion of dietary polyphenols promotes the proliferation of SCFA-producing bacteria such as *Bifidobacteria* and decreases the ratio of *Firmicutes* to *Bacteroidetes*. Polyphenols can alter the composition of the microbiota, thus determining changes in polyphenol metabolism and bioavailability, and bioactive metabolites can have beneficial effects on the host’s health (Table 4).

Figure 3 shows that the conversion of dietary polyphenols promotes the proliferation of SCFA-producing bacteria such as *Bifidobacteria* and decreases the ratio of *Firmicutes* to *Bacteroidetes*. This bidirectional interaction also accounts for polyphenol’s antioxidant effects.

### 5.3. Effect of Polyunsaturated Fatty Acids (PUFAs) and Conjugated Linoleic Acid (CLA) on Gut Homeostasis

n-3 PUFAs are a major component of fish fat and are widely ingested through food or supplements; they are involved in many biochemical processes in the human body and are well-known modulators of SCFAs produced by the gut microbiome [107].

Before the incorporation of PUFAs into the membrane of the target tissues from systemic circulation, dietary fats are emulsified in the stomach before they enter the small intestine, where they are cleaved off to form free fatty acids and 2-monoacylglyceride. In the enterocytes, PUFAs are re-esterified to triacyl glycerides, and, upon incorporation into chylomicrons, are transferred to the lymph and blood circulation. The concentrations of CLA and n-3 PUFAs in human blood plasma are markedly different—8 nm and 20 μM, respectively—but their significant increase can be induced by the dietary supplementation of CLAmix or fish oil [131,132]. Notably, since the biological activities of these PUFAs depend on their incorporation and metabolism in the target organ, the marked differences observed in the blood may not reflect the efficacy of their biological action [133].

Dietary lipids have been recognized to alter the gut microbiota composition by acting as substrates for bacterial metabolic processes or by modulating the growth of propionate- and butyrate-producing bacteria (e.g., *Bacteroides*, *Clostridium*, and *Roseburia*) [134]. A recent review article reported that the concentration of n-3 PUFAs in the blood is positively correlated with the abundance of human gut microbes, thus indicating that n-3 PUFAs could directly modulate the diversity and the abundance of gut microbiota [135]. In particular, dietary n-3 PUFAs regulate gut microbiota homeostasis, and their deficiency may induce dysbiosis [136]. Several clinical data indicate the ability of n-3 PUFAs to restore gut eubiosis in aged people or in those with several pathological conditions by increasing the abundance of butyrate-producing bacteria [137,138]. Other in vivo studies have reported that dietary n-3 PUFAs can increase the growth of lipopolysaccharide (LPS)-suppressing bacteria (*Bifidobacteria*) and limit that of LPS-producing bacteria (*Proteobacteria*) [139], whose presence can determine acute inflammatory responses by triggering the release pro-inflammatory cytokines. Interestingly, dietary lipids also have a relevant role in neural development, the differentiation of nerve cells, and the plasticity of the nervous system. Their importance is supported by in vivo and clinical trial studies that clearly show the crucial role of dietary n-3 PUFAs in brain development, ageing, and neurodegeneration (Table 5).

Conjugated Linoleic Acid (C18:2, CLA) is another PUFA that is attracting scientific interest due to its multiple beneficial effects, which, similarly to n-3 PUFAs, may be independent or dependent on the modulation of gut microbiota metabolism. The term CLA is the collective name generally used to indicate a further subclass of PUFAs without a methyl group between adjacent double bonds (conjugated diene of Linoleic Acid C18:2, LA). In particular, this generally refers to geometric and positional isomers of LA mainly present in dairy products and meat from ruminants (cis9, trans11 and trans10, cis12); the concentration of the former isomer typically ranges from 3 to 7 mg/g of fat. Commercially available dietary CLA supplement (CLAmix) is composed of an equimolar mixture of these two isomers. Dietary supplementation with the trans10, cis12 isomer is associated with the modulation of lipid metabolism and glucose tolerance, while the cis9, trans11 isomer has preeminent anti-oxidant and anti-inflammatory effects. The intake of an equimolar mixture of the two CLA isomers was suggested to effectively activate many biological pathways owing to the combined action of its isomers [149]. Moreover, dietary trans10, cis12 CLA exhibits the ability to increase the population of SCFA-producing bacteria and the cecal concentration of SCFAs (isobutyrate, acetate, and propionate) [150]. Similarly, trans10, cis12 CLA supplementation in a mouse model of obesity increased the level of butyrate-producing bacteria (*Butyrivibrio*, *Roseburia*, and *Lactobacillus*), leading to a significant increase in butyrate in the feces and in acetate in the plasma [151]. In this context, owing to the activity of the gut–brain axis, dysbiosis has been associated with the onset and progression of several neurological disorders [152]. In addition, the dysregulation of SCFA production has been linked to psychiatric illnesses and immune, metabolic, and neurodegenerative diseases [153].

In other studies, dietary LA may be indeed metabolized by several gut bacteria and converted into cis9, trans11 or some of its precursors (trans-11-18:1 or 10-hydroxy-18:1; 10-Oxo-trans-11-octadecenoic) [154]. These C18 fatty acids, similarly to SCFAs, exhibit the ability to activate the Nrf2 pathway [155,156,157,158]. Interestingly, some CLA by-products—such as 10-hydroxy-cis-12-octadecenoic acid (HYA) generated during Linoleic Acid metabolism—exhibit anti-inflammatory and antioxidant effects likely through GPR120-dependent activation [159,160] (Figure 3). Based on these data, it can be stated that the involvement of specific receptors/transporters in the CLA-mediated activation of Nrf2 needs further investigation. The possibility that CLA metabolites generated by gut microbiota may contribute to the biological activity of dietary CLA has been hypothesized [161]. However, different results have been presented, and the independence between the anti-inflammatory effect of CLA and the activity of the intestinal microbiota has recently been demonstrated [162].

## 6. Conclusions

In this review, we summarized the antioxidant mechanisms underlying dietary MACs, polyphenols, and PUFAs. The literature data reported in the field suggest that these molecules can exert their modulatory activity on the human redox status by acting in two main different ways: direct or indirect.

In particular, the antioxidant activity exhibited by PUFAs may be, at least in part, a consequence of their direct ability to target the Nrf2 pathway. In addition, based on the role of SCFAs as Nrf2 ligands, dietary MACs, polyphenols, and PUFAs may also have prebiotic activity, favoring the growth of SCFA-producing microbial populations, which are characteristic of eubiosis conditions.

Polyphenols and PUFA metabolites also exhibit an indirect effect, in which their biological activities can contribute to or enhance the effects triggered by the parent ingested molecules.

The key modulatory role played by SCFAs and polyphenol metabolites on the physiology of the host is mirrored by the network of molecular mechanisms underlying their direct activities, involving Nrf2 and HDAC signaling pathways and GPRs for SCFAs. Such complexity is further accentuated by the reciprocal interactions between these pathways.

SCFA production also represents important crossroads of the biological activities exerted by polyphenols and PUFAs.

The interplay among n-3 PUFA and CLA and gut microbiota is herein summarized for the first time, and the evidence of their direct and indirect actions on the physiology of the host likely contributes to expanding the nutritional and therapeutic importance of these dietary molecules.

In conclusion, the shared molecular pathways activated by polyphenols, PUFAs, and SCFAs support the role of Nrf2, HDACs, and GPRs as pharmacological targets. Since the activation of these pathways triggers a downstream signaling cascade, this can explain why dietary bioactive molecules can exert antioxidant/beneficial effects even when present in a low plasma concentration. A better understanding of the mechanisms underlying the prebiotic effects of polyphenols and PUFAs is an issue to be further explored that will shed light on the knowledge of their beneficial effects on human health.

## Figures and Tables

**Figure 1 antioxidants-12-01073-f001:**
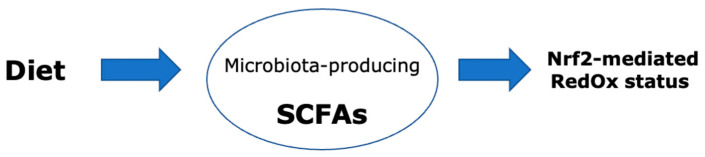
Possible role of SCFAs in the modulation of Nrf2-mediated redox homeostasis.

**Figure 2 antioxidants-12-01073-f002:**
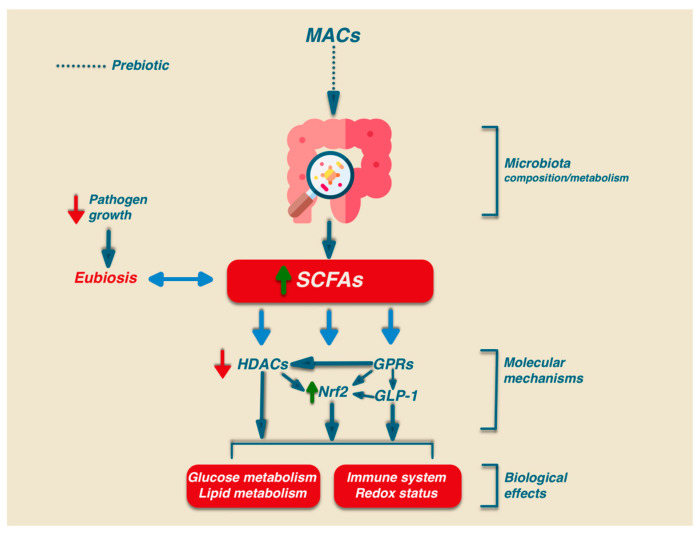
Biological effects of MACs. The intake of MACs can modulate the composition and metabolism of the microbiota, leading to an increased production of SCFAs. SCFAs exert biological effects by modulating specific signaling pathways. HDACs: histone deacetylases; GPCRs: G-protein-coupled receptors; GLP-1: glucagon like peptide 1; Nrf2: Nuclear factor 2-related erythroid 2.

**Figure 3 antioxidants-12-01073-f003:**
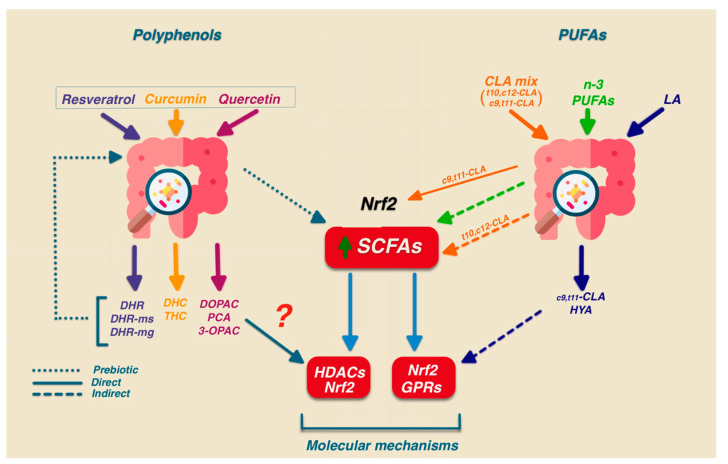
Antioxidant effects of polyphenols and PUFAs. Polyphenols and PUFAs can act indirectly or directly. Indirect effects act by favoring the growth of a specific microbial population, while direct effects affect the physiology of the host by modulating the Nrf2 pathway. DHR: Dihydroresveratrol DHR-ms: Dihydroresveratrol monosulfate; DHR-mg: Dihydroresveratrol monoglucuronide forms; DHC: Dihydrocurcumin; THC: tetrahydro curcumin; DOPAC 3,4-Dihydroxyphenylacetic acid; PCA: 3,4-dihydroxybenzoic acid; 3-OPAC: 3-hydroxyphenylacetic acid; HYA: 10-hydroxy-cis-12-octadecenoic acid.

**Table 1 antioxidants-12-01073-t001:** Studies reporting the link between gut dysbiosis/production of SCFAs in several human diseases. An increase or decrease in the levels considered is indicated by upward (↑) or downward arrow (↓), respectively.

Disease		Model	Microbiota Alteration Production of SCFAs	Ref.
**Diabetes**		Randomized clinical trial High-fiber diet	Type 2 diabetes ↓ SCFAs High fiber intake ↑ SCFAs ↑ SCFA-producing bacteria	[28]
		Meta analysis Dietary fiber	↑ Butyrate, propionate ↑ *Bifidobacterium*	[29]
**Inflammatory Bowel Disease (IBD)**		313 patients	↓ Acetate-to-butyrate converter *Firmicutes* (*Roseburia*) ↓ Propionate ↑ Pathogens (*Enterobacteriaceae*, *Proteobacteria*)	[30]
		127 patients 87 healthy controls	↓ Butyrate-producing bacteria (*Firmicutes*) ↓ SCFAs (acetate, propionate, butyrate)	[31]
		10 inactive Crohn patients 10 healthy controls	↓ SCFA-producing bacteria ↓ *Roseburia inulinivorans*, ↓ *Ruminococcus torques*, ↓ *Clostridium lavalense*, ↓ *Bacteroides uniformis* ↓ *Faecalibacterium prausnitzii*	[32]
**Nonalcoholic Fatty Liver** **Disease**		14 nonalcoholic fatty liver, 18 nonalcoholic steatohepatitis 27 healthy controls	↑ SCFA levels ↑ SCFA-producing bacteria (*Fusobacteriaceae*, *Prevotellaceae*)	[33]
		25 nonalcoholic fatty liver 25 nonalcoholic steatohepatitis 25 healthy donors	↓ *Ruminococcaceae* ↓ *Clostridiales* ↑ *Bacteroidetes* ↓ *Firmicutes*	[34]
		30 patients F0/1 fibrosis stage 27 patients F ≥ 2 fibrosis	↑ *Bacteroidetes* (F ≥ 2) ↑ *Ruminococcus* (F ≥ 2) ↓ *Prevotella*	[35]
**Neurodegeneration**	**Parkinson’s Disease**	Nonparametric meta-analysis	↑ *Akkermansia* ↓ Fecal SCFAs (acetate, propionate, butyrate)	[36]
		96 patients 85 controls	↓ Fecal SCFAs ↑ Plasma SCFAs ↑ Pro-inflammatory bacteria	[37]
		95 patients 33 controls	↓ Fecal SCFAs (propionic acetic, butyric) ↑ Plasma SCFA (propionic acetic)	[38]
	**Alzheimer’s Disease**	25 patients	↓ *Firmicutes*, *Bifidobacterium* ↑ *Bacteroidetes*	[39]
		33 dementia 22 mild cognitive impairment 120 subjective cognitive decline	↓ SCFA-producing bacteria (*Ruminococcus*, *Eubacterium*) ↑ AD biomarkers (Amyloid-β1-42 and p-tau concentrations)	[40]
		Mouse model Sodium butyrate supplementation	↓ Amyloid-β1-42 protein (40%)	[41]

**Table 2 antioxidants-12-01073-t002:** A brief summary of SCFAs produced by the gut microbial population and of their response to different receptors; adapted from [26,72]. Low or high affinity is denoted by + or ++, respectively.

Phylum	Family	Genus		FFAR3 (GPR41)	FFAR2 (GPR43)	GPR109A
**Firmicutes**	*Lachnospiraceae*	*Coprococcus*	**ACETATE**	**+ +**	**+ +**	**+ +**
		*Barnesiella*				
	*Ruminococcaceae*					
		*Akkermansia*				
		*Prevotella*				
		*Bifidobacterium*				
**Bacteroidetes**	*Bacteroidaceae*	*Bacteroides*	**PROPIONATE**	**+**	**+ +**	**+**
	*Prevotellaceae*	*Prevotella*				
	*Rikenellaceae*	*Alistipes*				
**Firmicutes**		*Eubacterium*				
		*Blautia*				
		*Coprococcus*				
	*Veillonellaceae*	*Dialister*				
	*Acidaminococcaceae*	*Phascolarctobacterium*				
**Verrucomicrobia**	*Verrucomicrobiaceae*	*Akkermansia*				
**Firmicutes**	*Lachnospiraceae*	*Eubacterium*	**BUTYRATE**	**+ +**	**+ +**	**+ +**
		*Roseburia*				
		*Clostridium*				
		*Eubacterium*				
		*Anaerostipes*				
		*Coprococcus*				
	*Ruminococcaceae*	*Faecalibacterium*				
		*Subdoligranulum*				
	*Erysipelotrichaceae*	*Holdemanella*				

**Table 4 antioxidants-12-01073-t004:** Effects of the dietary supplementation of polyphenols on the composition of microbiota. An increase or decrease in the levels considered is indicated by upward (↑) or downward arrow (↓), respectively.

Component	Animal Model	Effect on Gut Microbes	Ref.
Astaxanthin	β-carotene oxygenase 2 knockout mice	↑ *Mucispirillum schaedleri*, *Akkermansia*, *Muciniphila*	[115]
Fucoxanthin	azoxymethane/dextran sulfate sodium treated mice	↑ *Lachnospiraceae*, ↓ *Bacteroidlales*, *Rikenellaceae*	[116]
Tomato powder	BCO1/BCO2 double knockout mice	↑ *Lactobacillus*, *Bifidobacterium*, ↓ *Bacteroides*, *Mucispirillum*	[117]
Apple polyphenol extract	Wild-type mice	↑ *Verrucomicrobia*, *Akkermansia*	[118]
Blueberry extract	Sprague–Dawley rats	↑ Diversity of gut microbes	[119]
Curcumin	Wild-type mice	↑ *Akkermansia*, *Roseburia*, *Coprococcus*	[120]
	Wild-type mice	↑ *Muribaculaceae*, ↓ *Bacteroides*, *Ruminococcaceae*	[121]
Epicatechin gallate	Obese diabetic mice	↑ *Firmicutes: Bacteroidetes* ratio, ↑ *Lactobacillius*	[122]
Fu brick tea	Donor rats	↑ *Akkermansia maciniphilla*, *Bacteroides*, *Alloprevotella*	[123]
*Litchi chinensis* seed extract	Zebrafish	↑ *Trichococcus*, *Muribaculaceae*, *Lactobacillus*, ↓ *Micrococcaceae*, *Staphyllococcus*	[124]
Peanut skin procyanidin	DSS-induced ulcerative colitis in mice	↑ *Lachnospiraceae*, *Roseburia,* ↓ *Bacteroides*, *Helicobacter*, *Parabacteroides*	[125]
Pomegranate fruit pulp	Wild-type mice	↑ *Akkermansia maciniphilla*, *Parabacteroides distsonis*, *Bacteroides acidifaciens*	[126]
Purple sweet potato anthocyanin extract	Obese mice	↑ *Lactobacillus*, *Bifidobacterium*, *Akkermansia*	[127]
Tea polyphenols and polysaccharides	DSS-induced colitis in mice	↑ *Lactobacillus*, ↓ *Proteobacteria*, *Enterobacteriaceae*	[128]
*Triadica cochinchinensis* honey polyphenol	Cefixime-treated mice	↓ *Firmicutes/Bacteroidetes*	[129]
Xanthohumol	Male Tac: SW mice	↓ *Porphyromonadaceae*, *Lachnospiraceae*, *Lactobacillaceae*, ↑ *A. muciniphila*, *P. goldsteinii*, *A. finegoldii*	[130]

**Table 5 antioxidants-12-01073-t005:** Effects of PUFA dietary supplementation on microbiota composition and SCFA production. An increase or decrease in the levels considered is indicated by upward (↑) or downward arrow (↓), respectively.

	Fatty Acids		Effect on Gut Microbes	Ref.
**Clinical studies**	Omega-3 rich diet	A 45-year-old male consuming 600 mg of omega-3 (daily for 14 days)	↓ Species diversity ↑ Butyrate-producing bacteria (*Eubacterium*, *Roseburia*, *Anaerostipes*, *Coprococcus*, *Subdoligranulum*, *Pseudobutyrivibrio*)	[138]
	Enteral supplementation of a fish and safflower blended oil	32 premature infants with enterostomy (10 weeks)	↓ pathogenic bacteria (*Streptococcus*, *Clostridium*, *Escherichia*, *Pantoea*, *Serratia*, and *Citrobacter* genera)	[140]
	Omega-3 rich diet	Pregnant women	↑ *F. prausnitzii* species of the *Firmicutes* phylum ↓ *Bacteroides* genus of the *Bacteroidetes* phylum	[141]
	DHA/EPA	20 Healthy volunteers (8 wks, 4 g/day)	↑ SCFA-producing bacteria (*Bifidobacterium*, *Roseburia lactobacillus*)	[142]
	Estimated food intake of omega-3 fatty acids	876 female twins	↑ n3-PUFA ↑ SCFA-producing bacteria (*Lachnospiraceae* family)	[137]
	Omega-3 (sardines) (~3 g of EPA + DHA)	32 patients with type 2 diabetes 100 g of sardines (5 days per week for 6 months)	↓ *Firmicutes*/*Bacteroidetes* ratio, ↑ *Prevotella* genus in the omega-3 group	[143]
**Animal models**	n-3 PUFA	male C57BL/6 mice n-3 supplemented (n3+) n-3 deficient (n3−) vs control (CONT)	(n3−) ↓ SCFAs vs. CONT (n3+) ↓ Butyrate vs. CONT	[136]
	EPA-DHA	HFD-induced obese mice + EPA-DHA	↑ *Firmicutes*	[144]
	PUFAs omega-6 (n6) omega-3 (n3)	Wild-type mice fed + n3 or n6/(14 wks)	↓ proportion of *Bacteroidetes* phylum	[145]
	palm oil (PO), olive oil (OO) flaxseed/fish oil (FOO) compared with mice fed a low-fat diet (LF)	C57BL/6J mice fed with High-fat diet (HF+ PO, OO or FOO) compared with mice fed LF	HF+PO ↓ *Bacteroidetes* comp. to HF+OO HF+FFO ↑ Bifidobacterium comp. to LF	[146]
	High-fat diet (45%) with fish oil (FO) or lard (L)	C57Bl/6 Wild-type germ-free mice	FO ↑ *Lactobacillus* genus and *Akkermansia muciniphila* sp. L ↑ *Bilophila* genus of the *Proteobacteri* phylum	[147]
	Omega-3 PUFAs	male Sprague–Dawley rats	↑ *Bifidobacteria*	[148]

## Data Availability

Not applicable.

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
