# Peer review of "The Antioxidant Effect of Dietary Bioactives Arises from the Interplay between the Physiology of the Host and the Gut Microbiota: Involvement of Short-Chain Fatty Acids"

_antioxidants, 2023, doi:10.3390/antiox12051073_

Round 1
Reviewer 1 Report
The manuscript touches on very important issues concerning the antioxidant mechanisms underlying dietary MACs, polyphenols, and PUFAs. The publication is interesting and contains a lot of important information. Title clearly describes what the manuscript is about. Abstract adequately describes the work. Data of literature (137 manuscripts) are properly analyzed and interpreted. Most of the literature is from the last 10 years. Cited references not always correct. The information about the figures and tables aren’t correct in the work. The text contains many editorial errors and repetitions.
Detailed comments:
L.60-66: information on literature sources is missing. Please provide the reference.
L.4, L.81, : Dots should be remove.
L.143-146: information on literature sources is missing. Please provide the reference.
L.260: „the Nrf2 pathway is shown in Table II.” – where are Table II?
L.332-334: information on literature sources is missing. Please provide the reference.
L.402: “SCFAs production by bacteria_[102,103] may also” “_” should be deleted
L.407: „(Fu Y, et al., 2021).”??
L.415-417: information on literature sources is missing. Please provide the reference.
Figures and Tables:
L.261.: „Table 2009. 2017), Rey F.E., et al., (2010), González-Bosch et al., 2021” – Is it the title of the table?
„Tabale I” should be replaced with: „Table 1.”
„Table III” should be replaced with: „Table 3”
L.102: Scheme 1. Possible role of SCFAs in modulation of Nrf2-mediated redox homeostasis” should be replaced with: „Figure 1. Possible role of SCFAs in modulation of Nrf2-mediated redox homeostasis”
The abbreviations (below the Tables and Figures) should be explained.
Regarding Conclusion: This part is too long - should be significantly shortened.
Author Response
Reviewer 1.
The manuscript touches on very important issues concerning the antioxidant mechanisms underlying dietary MACs, polyphenols, and PUFAs. The publication is interesting and contains a lot of important information. Title clearly describes what the manuscript is about. Abstract adequately describes the work. Data of literature (137 manuscripts) are properly analyzed and interpreted. Most of the literature is from the last 10 years. Cited references not always correct. The information about the figures and tables aren’t correct in the work. The text contains many editorial errors and repetitions.
Detailed comments:
L.60-66: information on literature sources is missing. Please provide the reference.
Answer: The reference has been inserted (number 5)
L.4, L.81, : Dots should be removed.
Answer: Dots have been removed
L.143-146: information on literature sources is missing.Please provide the reference.
Answer: The reference has been inserted (number 25)
L.260: „the Nrf2 pathway is shown in Table II.” – where are Table II?
Answer: Table II has been corrected in Table 2
L.332-334: information on literature sources is missing. Please provide the reference.
Answer: The reference has been inserted (number 101)
L.402: “SCFAs production by bacteria_[102,103] may also” “_” should be deleted
Answer: the “_” has been deleted
L.407: „(Fu Y, et al., 2021).”??
Answer: (Fu Y, et al., 2021) has been substituted by number (135)
L.415-417: information on literature sources is missing. Please provide the reference.
Answer: The reference has been inserted (number 133)
Figures and Tables:
L.261.: „Table 2009. 2017), Rey F.E., et al., (2010), González-Bosch et al., 2021” – Is it the title of the table?
Answer: The references have been removed from the Table title
„Table I” should be replaced with: „Table 1.”
„Table III” should be replaced with: „Table 3”
Answer: All the table’s numbering has been corrected
L.102: Scheme 1. Possible role of SCFAs in modulation of Nrf2-mediated redox homeostasis” should be replaced with: „Figure 1. Possible role of SCFAs in modulation of Nrf2-mediated redox homeostasis”
The abbreviations (below the Tables and Figures) should be explained.
Answer: Scheme 1 has been changed in Figure 1 and all the abbreviations have been explained
Regarding Conclusion:This part is too long - should be significantly shortened.
Answer: The Conclusion section has been shortened

Reviewer 2 Report
This review manuscript described antioxidant dietary bioactive molecules especially short-chain fatty acids and how they link host and gut microbiota. The illustrations involved amounts of related studies and are well organized. Although there have been lots of reports demonstrating correlation of hot and gut microbiome, and SCFAs are established important metabolites between host and microbiome, this review paper summarized this correlation from a different perspective, from the antioxidant aspect. Therefore, this manuscript can still be a good reference for further antioxidant and related mechanism studies.
The major concern is that in part 5, unlike description of MAC, narration of Polyphenols and PUFAs/CLA are not focusing on their relation to gut microbiome and SCFAs. Tables clearly listing effects of MAC, PUFAs/CLA on gut microbiome and SCFAs can be included.
Other minor modifications needed are:
- Line 167-170 on page 5, looks like comments from your previous version.
- Double check the title of Table 2.
- Please double checks spelling/misunderstanding words, like than or that in line 309, probiotic or prebiotic in line 482?
Author Response
Reviewer 2.
This review manuscript described antioxidant dietary bioactive molecules especially short-chain fatty acids and how they link host and gut microbiota. The illustrations involved amounts of related studies and are well organized. Although there have been lots of reports demonstrating correlation of hot and gut microbiome, and SCFAs are established important metabolites between host and microbiome, this review paper summarized this correlation from a different perspective, from the antioxidant aspect. Therefore, this manuscript can still be a good reference for further antioxidant and related mechanism studies.
The major concern is that in part 5, unlike description of MAC, narration of Polyphenols and PUFAs/CLA are not focusing on their relation to gut microbiome and SCFAs. Tables clearly listing effects of MAC, PUFAs/CLA on gut microbiome and SCFAs can be included.
Answer: In the revised version we have inserted 2 new tables: Table 4 for Polyphenols & microbiota and Table 5 for PUFA & microbiota
Other minor modifications needed are:
Line 167-170 on page 5, looks like comments from your previous version.
Double check the title of Table 2.
Answer: In the revised version the title and the legend of Table 2 have been inserted
Please double checks spelling/misunderstanding words, like than or that in line 309, probiotic or prebiotic in line 482?
Answer: all the spelling/misunderstanding words have been checked

Reviewer 3 Report
In this review, the authors have attempted to compile some effects of dietary carbohydrates, polyphenols and PUFAs on their ability to directly or indirectly modulate the gut microbiota.
Some aspects that the authors should take into account are indicated.:
-The title seems to be not quite adequate for what the text is really about: the authors should reconsider another title.
- The authors should go more deeply into the subject of the review, as there are many assertions that are quite basic and are repeated throughout the manuscript. On the other hand, by contrast, there are many interesting aspects in the review that are simply referenced but not explored in depth.
-I. e., Paragraph 5.2 Polyfenols: rewrite the whole paragraph, there are ideas that are continually repeated in it (the same for other sections)
-Line 165: Salmonella (not S.)
- REFERENCES: there are some full names of magazines, write them in abbreviations.
Author Response
Reviewer 3.
In this review, the authors have attempted to compile some effects of dietary carbohydrates, polyphenols and PUFAs on their ability to directly or indirectly modulate the gut microbiota.
Some aspects that the authors should take into account are indicated.:
-The title seems to be not quite adequate for what the text is really about: the authors should reconsider another title.
Answer: The title has not been modified considering that all the other reviewers (especially 1 and 4) appreciated it.
- The authors should go more deeply into the subject of the review, as there are many assertions that are quite basic and are repeated throughout the manuscript. On the other hand, by contrast, there are many interesting aspects in the review that are simply referenced but not explored in depth.
-I. e., Paragraph 5.2 Polyfenols: rewrite the whole paragraph, there are ideas that are continually repeated in it (the same for other sections)
Answer: according to reviewers’ suggestions we have thoroughly revised the manuscript and in particular Paragraph 5.2 Polyphenols has been rewritten
-Line 165: Salmonella (not S.)
Answer: Salmonella has been corrected
- REFERENCES: there are some full names of magazines, write them in abbreviations.
Answer: all the references have been corrected
Reviewer 4 Report
The manuscript entitled “Antioxidant effect of dietary bioactives arise from the interplay between host and gut microbiota physiology: involvement of Short-Chain Fatty Acids” is a review article to discuss “the main mechanisms through which MACs, polyphenols and PUFAs can modulate host redox homeostasis via their ability to directly or indirectly activate the Nrf2 pathway”. Although the title is good and interesting, but the contents were not well organized and too complex to read. Besides, there are some problems in the manuscript needed to be solved before resubmitting.
Major revision:
1. The subtitles are not good or right for their contents.
2. The contents should be concise for important review.
3. It is very odd to see the suggestions below some sections, like: “Authors should discuss the results and how they can be interpreted from the per-spective of previous studies and of the working hypotheses. The findings and their impli- cations should be discussed in the broadest context possible. Future research directions may also be highlighted” or “This section is not mandatory but can be added to the manuscript if the discussion is unusually long or complex”.
4. The conclusions are too long.
Minor revision:
1. Table I should be put in the section 4.
2. Figure 1 should be in the section 5.1.
3. English should be polished by native English colleagues or publishers.
Author Response
Reviewer 4.
The manuscript entitled “Antioxidant effect of dietary bioactives arise from the interplay between host and gut microbiota physiology: involvement of Short-Chain Fatty Acids” is a review article to discuss “the main mechanisms through which MACs, polyphenols and PUFAs can modulate host redox homeostasis via their ability to directly or indirectly activate the Nrf2 pathway”.Although the title is good and interesting, but the contents were not well organized and too complex to read. Besides, there are some problems in the manuscript needed to be solved before resubmitting.
Major revision:
- The subtitles are not good or right for their contents.
Answer: All the subtitles have been modified
- The contents should be concise for important review.
Answer: according to the reviewers’ suggestions we have thoroughly revised the manuscript
- It is very odd to see the suggestions below some sections, like: “Authors should discuss the results and how they can be interpreted from the perspective of previous studies and of the working hypotheses. The findings and their implications should be discussed in the broadest context possible. Future research directions may also be highlighted” or “This section is not mandatory but can be added to the manuscript if the discussion is unusually long or complex”.
Answer: we apologize for the mistake. The comments inserted by mistake and coming from an earlier version of the manuscript have been eliminated.
- The conclusions are too long.
Answer: The Conclusion section has been shortened
Minor revision:
- Table I should be put in the section 4.
Answer: Table 1 has been moved in the section 4
2. Figure 1 should be in the section 5.1.
Answer: We moved Figure 1 after the Introduction since it is a schematic representation of the possible role of SCFAs in the modulation of Nrf2-mediated redox homeostasis.
3. English should be polished by native English colleagues or publishers.
Answer: the English have been revised by a native English
Reviewer 5 Report
The article concerns the role of bioactive fatty acids, manly SCFA, in probiosis. The views described are important and allow to better plan the prebiotic diet, especially important for people with dysbiosis. Therefore, it should be published with some suggested changes.
Italics should be used for all Latin names, e.g. Proteobacteria p. 2, line 56.
At the end of two paragraphs, there are sentences that seem to be a draft commentary for the preparation of the manuscript and should be deleted from the final version (Page 5, line 167-170: "Authors should discuss the results and how they can be interpreted from the perspective of previous studies and of the working hypotheses. The findings and their implications should be discussed in the broadest context possible. Future research directions may also be highlighted”; Page 6, line 212-213: “This section is not mandatory but can be added to the manuscript if the discussion is unusually long or complex").
Page 9, line 276-277: The sentence confuse single non-digestible polysaccharides with complex products, oat and wheat bran is a grain fraction, chemically complex, should not be referred to as non-digestible polysaccharides. These bran contain a number of non-digestible polysaccharides, such as beta-glucan. Only individual polysaccharide chemicals should be included in this sentence, not complex products.
Page 10, line 306: not all polyphenols have two phenolic rings, e.g. phenolic acids do not.
Page 10, lines 315-322: “Small polyphenols can be easily absorbed after de-glycosylation (does de-glycosylation occur under the influence of endogenous or bacterial enzymes?) and then they are converted in soluble metabolites through Phase I (oxidation, reduction , and hydrolysis) and Phase II reactions (conjugation). Complex polyphenols (oligomeric and polymeric) need to be transformed by specific gut enzymes to increase their bioavailability and circulating levels in the plasma. As glycosides, polyphenols are first converted into aglycones through specific enzymatic transformations, including C-ring cleavage, decarboxylation, dehydroxylation, and demethylation (please consider if the transformation to aglycones is based on de-glucosylation, under the influence of bacterial enzymes, and only after de-glucosylation, further reactions take place). The released aglycones, after absorption into the small intestine (whether in the small intestine or in later gut sections?), can be further converted into enterocytes and hepatocytes (polyphenols converted into cells?).”
P. 10, line 326-327: "Their concentrations in the plasma are higher in comparison with the ingested molecules." This sentence is unclear, what concentrations are compared?
Page 11, Figure 2 describes the effects of PUFAs, which are described far below. In accordance with the guidelines for including figures in the text, such a summary diagram should be placed only below the text that describes it, so it should be further, after subsection 5.3.
Author Response
Reviewer 5
The article concerns the role of bioactive fatty acids, manly SCFA, in probiosis. The views described are important and allow us to better plan the prebiotic diet, especially important for people with dysbiosis. Therefore, it should be published with some suggested changes.
Italics should be used for all Latin names, e.g. Proteobacteria p. 2, line 56.
Answer: The Latin names have been corrected
At the end of two paragraphs, there are sentences that seem to be a draft commentary for the preparation of the manuscript and should be deleted from the final version (Page 5, lines 167-170: "Authors should discuss the results and how they can be interpreted from the perspective of previous studies and of the working hypotheses. The findings and their implications should be discussed in the broadest context possible. Future research directions may also be highlighted”;
Page 6, lines 212-213: “This section is not mandatory but can be added to the manuscript if the discussion is unusually long or complex").
Answer: we apologize for the mistake. The comments inserted by mistake and coming from an earlier version of the manuscript have been eliminated.
Page 9, line 276-277: The sentence confuse single non-digestible polysaccharides with complex products, oat and wheat bran is a grain fraction, chemically complex, should not be referred to as non-digestible polysaccharides. These bran contain a number of non-digestible polysaccharides, such as beta-glucan. Only individual polysaccharide chemicals should be included in this sentence, not complex products.
Answer: We have corrected the sentence as suggested by the reviewer
Page 10, line 306: not all polyphenols have two phenolic rings, e.g. phenolic acids do not.
Answer: We have corrected the sentence inserting “at least one”
Page 10, lines 315-322: “Small polyphenols can be easily absorbed after de-glycosylation (does de-glycosylation occur under the influence of endogenous or bacterial enzymes?) and then they are converted in soluble metabolites through Phase I (oxidation, reduction , and hydrolysis) and Phase II reactions (conjugation). Complex polyphenols (oligomeric and polymeric) need to be transformed by specific gut enzymes to increase their bioavailability and circulating levels in the plasma. As glycosides, polyphenols are first converted into aglycones through specific enzymatic transformations, including C-ring cleavage, decarboxylation, dehydroxylation, and demethylation (please consider if the transformation to aglycones is based on de-glucosylation, under the influence of bacterial enzymes, and only after de-glucosylation, further reactions take place). The released aglycones, after absorption into the small intestine (whether in the small intestine or in later gut sections?), can be further converted into enterocytes and hepatocytes (polyphenols converted into cells?).”
Answer: We have corrected the sentence as suggested by the reviewer (page 9, lines 320-323)
And the sentence (page 9, lines 323-324)
The sentence ”the released aglycones, after absorption into the small intestine” is correct
Pag. 10, lines 326-327: "Their concentrations in the plasma are higher in comparison with the ingested molecules." This sentence is unclear, what concentrations are compared?
Answer: We have changed the sentence (page 9, lines 328-329)
Page 11, Figure 2 describes the effects of PUFAs, which are described far below. In accordance with the guidelines for including figures in the text, such a summary diagram should be placed only below the text that describes it, so it should be further, after subsection 5.3.
Answer The Figure 2 of the original version of the manuscript (now Figure 3) is placed after it is mentioned for the first time.

Round 2
Reviewer 3 Report
Re-review: The authors have made the effort to correct those parts that were confusing or needed further explanation. The manuscript is now fit for publication.
Reviewer 4 Report
Dear authors:
It is a comprehensive review after well organized.